# Amplified centrosomes—more than just a threat

Eva Kiermaier [1✉], Isabel Stötzel [1], Marina A Schapfl [2] & Andreas Villunger [2,3✉]

## Abstract

Centrosomes are major organizing components of the tubulin-based cytoskeleton. In recent years, we have gained extensive knowledge about their structure, biogenesis, and function from single cells, cell–cell interactions to tissue homeostasis, including their role in human diseases. Centrosome abnormalities are linked to, among others primary microcephaly, birth defects, ciliopathies, and tumorigenesis. Centrosome amplification, a state where two or more centrosomes are present in the G1 phase of the cell cycle, correlates in cancer with karyotype alterations, clinical aggressiveness, and lymph node metastasis. However, amplified centrosomes also appear in healthy tissues and, independent of their established role, in multi-ciliation. One example is the liver where hepatocytes carry amplified centrosomes owing to whole-genome duplication events during organogenesis. More recently, amplified centrosomes have been found in neuronal progenitors and several cell types of hematopoietic origin in which they enhance cellular effector functions. These findings suggest that extra centrosomes do not necessarily pose a risk for genome integrity and are harnessed for physiological processes. Here, we compare established and emerging 'non-canonical functions' of amplified centrosomes in cancerous and somatic cells and discuss their role in cellular physiology.

**Keywords** Centrosomes; Cancer; Migration; Differentiation; Immunity
**Subject Categories** Cancer; Cell Adhesion, Polarity & Cytoskeleton

## Introduction

The centrosome is a membrane-less organelle, which acts as the main organizer of the microtubule (MT) cytoskeleton in most animal cells. Owing to their capacity to grow MT filaments, they orchestrate various fundamental cellular processes, such as cell division, vesicle trafficking, cell polarization, motility, and ciliogenesis. During these processes, centrosomes direct the MT cytoskeleton to provide internal structural support, regulation of cell shape and polarity, as well as trafficking routes on which vesicular cargos are transported.

First described in 1876 by Édouard van Beneden, the centrosome attracted much attention due to its role in assembling the mitotic spindle, which separates the duplicated sister chromatids into the daughter cells. In 1887, Theodor Boveri's drawings of dividing sea-urchin eggs first associated centrosome abnormalities with tumorigenesis. In particular, increased centrosome numbers, or so-called centrosome amplification (CA), can interfere with the fidelity of chromosome segregation during mitosis and contribute to genetic instability, aneuploidy, and tumor progression. Since, amplified centrosomes have been identified in various solid tumors such as breast, prostate, colon, ovarian and pancreatic cancer (Lingle et al, 1998; Pihan et al, 1998; Hsu et al, 2005; Sato et al, 1999) as well as hematological malignancies (Krämer et al, 2005; Giehl et al, 2005). CA has also been implicated in contributing to chromosomal instability (CIN), metastasis, and poor clinical prognosis of cancer patients (Pihan et al, 2003, 1998; Sato et al, 1999; Krämer et al, 2005).

During the past decade, several studies have further investigated the molecular links between CA, CIN, and tumorigenesis. Boveri initially proposed that CA-mediated malignant tumors are the result '*of a certain abnormal chromosome constitution, which in some circumstances can be generated by multipolar mitoses*' (Boveri, 2008). This hypothesis was questioned by David Hansemann, who reported abnormal mitotic figures and distribution of chromosomes to daughter cells in carcinomas, but stated that asymmetric nuclear divisions can also occur in benign lesions, or during tissue overgrowth (Hardy and Zacharias, 2005).

Whether extra centrosomes are causative for tumorigenesis has been subject of discussion—indeed, CA is sufficient to induce cell transformation and cancer development in several model organisms (Basto et al, 2008; Crasta et al, 2012; Coelho et al, 2015; Serçin et al, 2016; Levine et al, 2017). However, in all of these studies, CA was either induced by overexpression of PLK4, a key regulator of centriole biogenesis (further discussed in the section "(De) Regulation of centrosome numbers"), or the consequences of CA have been studied on a *p53* mutant background. As PLK4 is also required for maintaining chromosomal stability and cell motility (Rosario et al, 2010, 2015), it is often difficult to rule out that tumorigenesis is associated with other roles of PLK4 than centrosome duplication. Moreover, high levels of CA can cause severe fitness defects incompatible with cell transformation (Levine et al, 2017; Kulukian et al, 2015; Vitre et al, 2015) or eventually even delay cancerogenesis (Braun et al, 2024). Lastly, CA and CIN are also prevalent in other non-cancer pathologies, including autosomal dominant polycystic kidney and liver disease where impaired ciliary-mediated intracellular signals promote cell hyperproliferation and cyst formation (Battini et al, 2008; Masyuk et al, 2014; Dionne et al, 2018) (Fig. 1A).

[1]Life and Medical Sciences Institute, Immune and Tumor Biology, University of Bonn, Bonn, Germany. [2]Institute for Developmental Immunology, Biocenter, Medical University of Innsbruck, Innsbruck, Austria. [3]The Research Center for Molecular Medicine (CeMM) of the Austrian Academy of Sciences, Lazarettgasse 14, 1090 Vienna, Austria. ✉E-mail: ekiermai@uni-bonn.de; andreas.villunger@i-med.ac.at

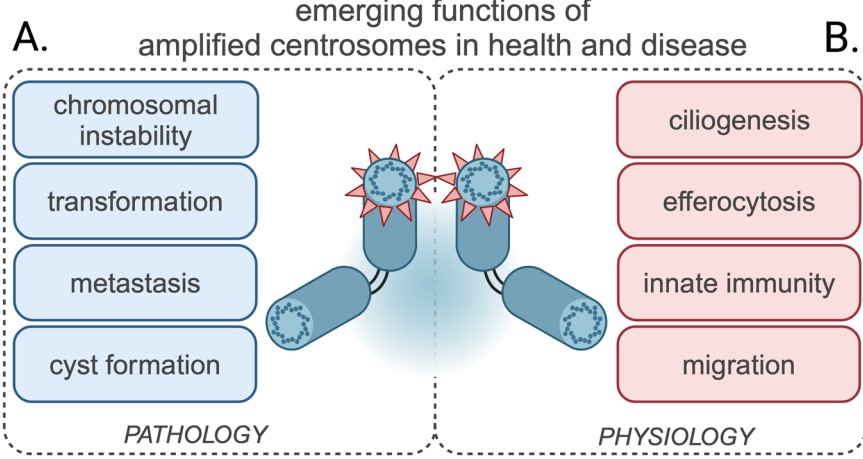

**Figure 1. Emerging functions of amplified centrosomes in health and disease.**

(A) Alterations in centrosome numbers can be associated with tumorigenesis and metastasis, as well as autosomal dominant polycystic kidney and liver disease (B) Amplified centrosomes can also appear naturally during terminal differentiation into highly specialized cell types, where they enhance specific effector functions such as ciliogenesis and innate immune processes thus questioning whether CA necessarily poses a risk to human health.

> **Box 1  In need of answers**
>
> Which developmental cues or stress-related signals lead to CA in diploid cells?
>
> Do we know all cell types that can amplify centrosomes to improve cell function?
>
> When do signals from extra centrosomes support physiology or pose a fitness risk?
>
> Does centrosome copy number define downstream signaling strength?
>
> Why do polyploid cells tolerate extra centrosomes and how do they impact physiology?

Studies in highly differentiated cell types, such as hepatocytes, olfactory sensory neurons, innate immune cells, osteoclasts, and megakaryocytes, in which amplified centrosomes support different cell functions linked to differentiation and tissue homeostasis, also support a more nuanced view on the phenomenon of CA and its functional consequences (Guidotti et al, 2003; Ching and Stearns, 2020; Weier et al, 2022; Philip et al, 2022; Becker et al, 2024) (Fig. 1B). These observations raise questions about the distinct roles of amplified centrosomes in healthy tissues and how the cell copes with additional centrosomes during division, for instance, in hepatocytes during liver regeneration. Together, these findings imply that the current view of amplified centrosomes as a unique and unambiguous hallmark of malignant disease has to be revised and raise the question why CA can be tolerated in certain cell types, but not in others (Box 1).

Here, we discuss the roles of amplified centrosomes focusing on development and differentiation of non-ciliated cells, as well as its relation to cell signaling, function, and tissue homeostasis. Centriole duplication in the context of multi-ciliation during differentiation and associated pathologies, and signaling events downstream of the spindle pole bodies (SPBs), which serve as centrosome equivalent in fungi, have been recently discussed in these excellent reviews (Breslow and Holland, 2019; Langlois-Lemay and D'Amours, 2022).

## Centrosome structure

The centrosome consists of a pair of cylindrical centrioles, which are connected through a proteinaceous linker and surrounded by the pericentriolar material (PCM) that contains the functional components for MT nucleation and organization (Bornens, 2012). The centrioles show a defined ultrastructure characterized by MT triplets arranged at a nine-fold radial symmetry, whereas the PCM was traditionally described as amorphous, electron-dense material that lacks a defined ultrastructure (reviewed by Banterle and Gönczy, 2016). Yet, advances in super-resolution light microscopy allowed the identification of concentric layers of PCM components that surround centrioles during interphase (Lawo et al, 2012; Mennella et al, 2012; Fu and Glover, 2012; Sonnen et al, 2012), indicating a higher-order organization of the PCM. Anchoring and nucleation of MTs is further mediated by γ-tubulin, which is localized in ring structures embedded within the PCM (Oakley et al, 1990; Stearns et al, 1991; Zheng et al, 1995; Moritz et al, 1998). These γ-tubulin ring complexes (γTuRCs) are the template for assembling MTs, which are anchored at the centrosome via their minus ends (reviewed by Kollman et al, 2011). γTuRCs first assemble within the cytoplasm and are further recruited to the centrosome via PCM proteins such as CDK5RAP2 and NEDD1 (Fong et al, 2008; Lüders et al, 2006; Manning et al, 2010). This process is most prominent at the transition from interphase to mitosis and accompanied by marked changes in PCM size and an increased ability to nucleate MT filaments. Of note, γ-tubulin-independent MT nucleation mechanism(s) have been described in Drosophila larval brain cells (Zhu et al, 2023). Here, centrosomes are still able to nucleate MTs in the absence of γTuRC, mediated by

the tumor-overexpressed gene (TOG) domain protein, Mini-spindles (*MSPS*).

In addition to centrosomes, several other non-centrosomal structures can act as MTOC, such as the Golgi apparatus, the nuclear envelope and the plasma membrane (reviewed by Muroyama and Lechler, 2017; Bartolini and Gundersen, 2006). Moreover, kinetochores and spindle MTs can establish centrosome-independent MT organization during mitosis (Maiato et al, 2004; Janson et al, 2005). This MT-dependent nucleation of MT filaments requires the Augmin complex to recruit γTuRC to the surface of pre-existing long-lived MTs (Goshima et al, 2007, 2008; David et al, 2019) and contributes to branching and amplification of MT numbers within the mitotic spindle (reviewed by Travis et al, 2022). MT nucleation from non-centrosomal sites often coincides with loss of centrosomal MTOC activity, which seems to be predominantly regulated by relocalization of PCM proteins to these sites (Yang and Feldman, 2015; Muroyama et al, 2016; Pimenta-Marques et al, 2016). Yet, the signals that induce such rearrangements remain poorly understood in many cell types.

## (De)Regulation of centrosome numbers

In dividing cells, centrosome numbers are tightly controlled during the cell cycle, and limited to one centrosome in G1 phase and two prior to mitosis. The centriole duplication cycle is further coupled to DNA replication ensuring timely synchronization of centriole and chromosome duplication (reviewed by Fırat-Karalar and Stearns, 2014; Banterle and Gönczy, 2016; Pereira et al, 2021). In G1 phase, cells contain one centrosome with exactly two centrioles, one older mother and one daughter centriole, which are tightly connected via a flexible linker structure that tethers the proximal ends of both centrioles (Bornens et al, 1987; Paintrand et al, 1992). Mother and daughter centrioles structurally differ from each other by distal and subdistal appendages that are only present on mature mother centrioles (Piel et al, 2000).

Centriole duplication starts at the G1-S transition, when a new procentriole grows orthogonally from the proximal end of the two existing parental centrioles. During S and G2 phase, procentrioles elongate and remain connected to the parental centrioles. At the end of G2, the linker between the two original pairs of centrioles dissolves, which is regulated by NEK2 kinase-mediated phosphorylation of linker proteins leading to their displacement (Mardin et al, 2010; Faragher and Fry, 2003; Fry et al, 1998). The duplicated centriole pairs, each consisting of a mother and a new daughter centriole, separate and form the poles of the mitotic spindle. After successful chromosome segregation and cytokinesis each new daughter cell contains one pair of centrioles, which disengage from each other and mature, thus allowing a new round of centriole duplication (reviewed by Nigg and Holland, 2018).

In mammals, centriole duplication is initiated by the Polo-like kinase 4 (PLK4) that accumulates at the proximal ends of the pre-existing centrioles followed by recruitment of centrosomal proteins, such as SIL/STIL, HsSAS-6, CPAP, CEP135, and CEP110, to form the base of the new procentriole (Bettencourt-Dias et al, 2005; Habedanck et al, 2005; Kleylein-Sohn et al, 2007). PLK4 is recruited to the parental centriole via binding to the adapter proteins CEP152 and/or CEP192 present within the PCM (Cizmecioglu et al, 2010; Hatch et al, 2010).

Two major mechanisms have been proposed that would restrict centrosome numbers. First, accumulation of defined levels of PLK4 at the base of the parental centriole determines the location of new centriole growth from the pre-existing mother centriole. The limited amount of PLK4 at the G1-S transition ensures that centriole growth can occur only once during the cell cycle. Consistent with this model, PLK4 initially localizes around the base of the pre-existing centriole in a ring-like pattern and concomitantly relocalizes to a single focus at the centriole wall (Ohta et al, 2014). This process involves auto-phosphorylation and intrinsic self-organization of PLK4 (Park et al, 2019; Yamamoto and Kitagawa, 2019). PLK4 auto-phosphorylation has also been demonstrated to regulate its own degradation by promoting ubiquitylation, thereby limiting PLK4 levels and maintaining a constant number of centrioles (Cunha-Ferreira et al, 2009; Rogers et al, 2009; Guderian et al, 2010; Cunha-Ferreira et al, 2013; Klebba et al, 2015). Recent structural studies using expansion microscopy further revealed that PLK4 localizes to discrete spots along the wall of parent centrioles, which directs procentriole formation (Scott et al, 2023). Importantly, PLK4 self-phosphorylation regulates the release of active PLK4 from the centriole wall and instructs a single site for procentriole growth.

A second mechanism to block reduplication and restrict centriole numbers operates centrosome-intrinsically. Cell fusion experiments revealed that only unduplicated centrioles from G1 phase were able to duplicate, while centrioles from a G2 cell were not (Wong and Stearns, 2003). Mechanistically, engagement of mother and procentrioles in G2 and M phase blocks reduplication of centrioles (Tsou and Stearns, 2006). Disengagement of centrioles at the M-G1 transition requires the activity of the mitotic kinase PLK1 and separase, a protease that cleaves cohesion rings at the metaphase-to-anaphase transition (Tsou et al, 2009; Schöckel et al, 2011). Centrosome duplication and DNA replication are further linked by the usage of overlapping proteins that ensure timely coordination and propagation of both centrosomal and DNA content. At the G1/S-transition, CDK2 localizes at the centrosome (Hinchcliffe et al, 1999; Meraldi et al, 1999; Lacey et al, 1999; Matsumoto et al, 1999) while entry into mitosis requires recruitment of CDK1 to centrosomes, which is mediated by the centrosomal protein CEP63 (Ferguson et al, 2010; Löffler et al, 2011). Moreover, CDK1 and the mitotic Cyclin B bind to centrosomal STIL and thereby hinder formation of the PLK4-STIL complex. This in turn prevents untimely phosphorylation of STIL by PLK4 and limits centriole biogenesis to only once per cell cycle (Zitouni et al, 2016).

Several mechanisms have been described to account for CA, such as dysregulation of the centrosome duplication cycle, mitotic defects or cell fusion events (Nigg, 2006; Cosenza and Krämer, 2016). Impaired regulation of the centriole duplication machinery can lead to overduplication of centrioles and excessive growth of procentrioles around one or both parental centrioles. Overduplication of centrioles is predominantly caused by altered expression levels of proteins involved in regular centriole duplication. In particular, overexpression of PLK4 results in overduplication of centrioles in various cell types and species (Bettencourt-Dias et al, 2005; Habedanck et al, 2005; Peel et al, 2007; Basto et al, 2008; Levine et al, 2017) further emphasizing a crucial role for regulating the levels of this initiator protein to ensure faithful centriole duplication.

Moreover, PLK4 expression is upregulated in human breast and colon cancers (Macmillan et al, 2001; Marina and Saavedra, 2014) but, somewhat surprisingly, correlates with low relapse-free survival in breast cancer patients (Jiawei et al, 2022). Nonetheless, targeting PLK4 has emerged as promising strategy for anti-cancer treatments (reviewed by Liu, 2015). In addition to PLK4, overexpression of other core centrosomal proteins, such as STIL and SAS6 as well as PCM components like pericentrin, can induce centrosome overduplication (Leidel et al, 2005; Strnad et al, 2007; Loncarek et al, 2008; Vulprecht et al, 2012).

A recent study identified over-elongation of centrioles in a variety of human cancer cell lines as another possible cause of CA (Marteil et al, 2018): overly long centrioles fragment after PLK4 inhibition, which can lead to CA in the absence of centriole biogenesis, confirmed at an ultra-structural level. Moreover, centriole over-elongation triggers ectopic procentriole formation along the elongated centriole wall (Marteil et al, 2018). What causes centriole over-elongation in cancer cells and which mechanisms regulate centriole length are still elusive and require further investigations.

In addition, perturbation of cell-cycle progression can result in defects in centriole duplication. A prolonged G2 arrest leads to PLK1 activation and has been shown to induce premature centriole disengagement and centriole reduplication in G2 phase (Lončarek et al, 2010; Dwivedi et al, 2023).

In contrast to centriole overduplication, mitotic defects can also lead to accumulation of centrosomes. As such, impaired or incomplete mitosis, caused by mitotic slippage, for instance, in response to chromosome alignment or segregation defects, or defective cytokinesis itself, cause not only an increase in DNA content (ploidy), but also in centrosome numbers (Meraldi et al, 2002; Fujiwara et al, 2005; Ganem et al, 2007; Duensing et al, 2008; Davoli and de Lange, 2012). Similarly, CA can also derive from endoreduplication or infection-induced cell–cell fusion events (Duelli et al, 2007). In most of these cases, CA negatively impacts cellular fitness, raising the question about checkpoints that safeguard or re-establish correct centrosome number.

# Signaling events elicited by (amplified) centrosomes

## Connecting extra centrosomes with p53 signaling

It is now well established that alterations in centrosome numbers—both their increase and loss—can result in activation of the p53 pathway (Lambrus et al, 2016; Meitinger et al, 2016; Fong et al, 2016; Fava et al, 2017). While centrosome loss triggers p53 activation indirectly, due to mitotic delays that ultimately engage the mitotic surveillance pathway (aka STOP watch pathway) (reviewed by Phan and Holland, 2021), extra mature centrosomes are sensed by a different mechanism, leading to the formation of the PIDDosome. This multi-protein complex, consisting of PIDD1 and RAIDD/CRADD, recruits the pro-enzymatic form of a cysteine-directed and aspartate-specific endopeptidase, Caspase-2, to engage p53, ultimately limiting cell proliferation (Fava et al, 2017) (Fig. 2A).

PIDD1, initially identified as a p53 target gene and implicated along with Caspase-2 in cell death induced by genotoxic stress (Lin

et al, 2000; Tinel and Tschopp, 2004; Weiler et al, 2022), localizes to the mother centriole in healthy cells via interaction with the distal appendix protein, ANKRD26 (Evans et al, 2021; Burigotto et al, 2021). Accumulation of an additional mature centriole, which was for instance observed after cytokinesis failure, suffices to trigger PIDDosome formation. CA induced by PLK4 overexpression causes a similar response, excluding increases in cellular ploidy as a cause for pathway activation (Fava et al, 2017). Complex formation results in activation of Caspase-2 that in turn processes and neutralizes the major regulator of p53 protein accumulation, the E3 ligase MDM2 (Fig. 2A). As a consequence, PIDDosome activation leads to p53-dependent and p21-mediated cell cycle arrest in epithelial cancer cell lines accumulating extra centrosomes (Fava et al, 2017). Whether p53 signaling strength or duration depends on the number of amplified centrosomes per cell has not been assessed. What is clear though is that mother centrioles need to cluster in order to trigger signaling, as interfering with MT dynamics abrogates clustering and p53 activation (Burigotto et al, 2021). This anti-proliferative response may aid the restoration of a normal centrosome number found upon genetically induced CA in model cell lines due to the competitive disadvantage and reduced growth potential of such cells (Sala et al, 2020; Bloomfield and Cimini, 2023).

## Centrosomes as inducers of cell-death signaling

Recently, the centrosome has taken centerstage as a signaling hub in the context of bacterial infection-driven cell death of macrophages where it aids the assembly of the NLRP3 inflammasome, a multi-protein complex similar to the PIDDosome (Fig. 2B). NLRP3 and the adapter protein ASC (apoptosis-associated speck-like protein containing a CARD) are required for the activation of Caspase-1, a process facilitated by PLK4-mediated phosphorylation of NEK7. This in turn favors inflammasome assembly, maturation of inflammatory cytokines and pyroptosis (He et al, 2016; Schmid-Burgk et al, 2016; Magupalli et al, 2020), a lytic form of cell death, initiated by proteolytically activating the pore-forming protein Gasdermin D, GSDMD (Shi et al, 2015).

Whether a similar pathway can operate downstream of centrosomes in other somatic or transformed cells to limit CA has not been reported, yet cell death was observed in cells amplifying centrosomes (Braun et al, 2024; Bloomfield and Cimini, 2023).

Current evidence suggests that CA, for instance induced by *Plk4* overexpression or cytokinesis failure, engages the mitochondrial apoptosis pathway: both normal and transformed white blood cells are highly susceptible to apoptosis in response to *Plk4* overexpression or cytokinesis failure. Loss of individual PIDDosome components or overexpression of BCL2, an inhibitor of mitochondrial apoptosis, preserves cell survival in cells with CA (Braun et al, 2024). A study on bioRxiv provides a potential explanation how the PIDDosome can trigger the activation of apoptosis effectors at mitochondria in response to extra centrosomes. The pro-apoptotic BCL2 family protein BID serves as a Caspase-2 substrate, becoming activated by proteolysis to promote activation of the cell death effectors BAX/BAK, best seen in human blood cancer cell lines (Rizzotto et al, 2024). This study would explain how inhibition of Aurora B kinase, leading to cytokinesis failure and CA, can also kill p53-deficient cancer cells (Sun et al, 2014). Here, BID processing abrogates the need for p53 pathway activation to promote

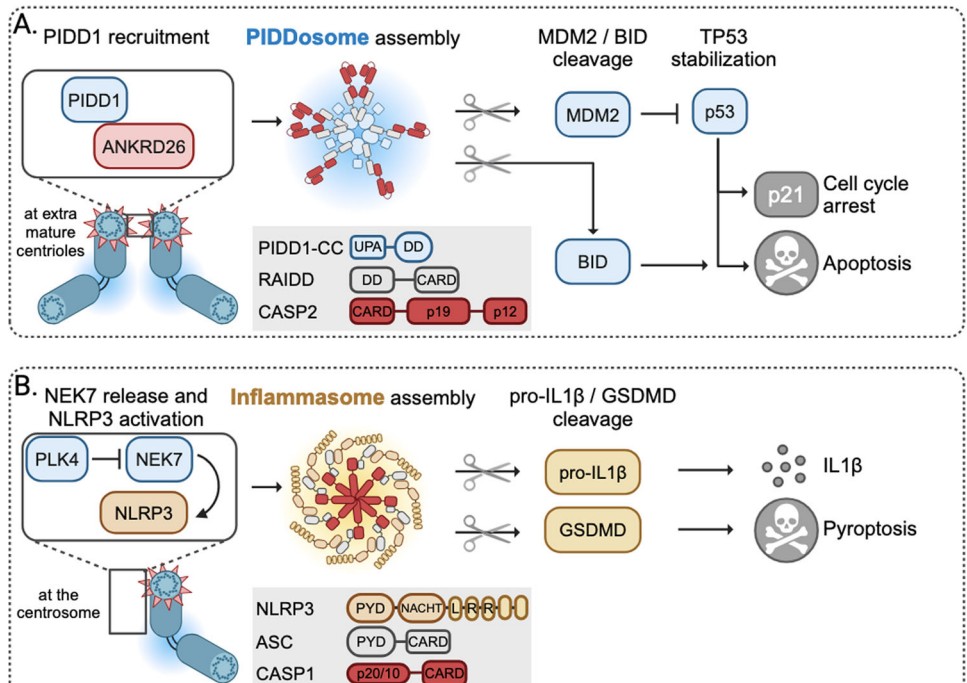

**Figure 2. Signaling events downstream of extra centrosomes.**

(A) Cycling cells typically contain a pair of centrioles, consisting of one daughter and one mature parent centriole decorated with distal appendages (indicated in red). The presence of extra mature centrioles leads to PIDD1 recruitment via ANKRD26 and its activation. The oligomerization of PIDD1, RAIDD and caspase-2 facilitates PIDDosome assembly and activates caspase-2. Caspase-2 cleaves either pro-apoptotic protein BID or MDM2 causing P53 stabilization and subsequent cell cycle arrest or apoptotic cell death. (B) Deubiquitination of PLK4 facilitates phosphorylation of NEK7, which in turn reduces NEK7-NLRP3 inflammasome interaction. NLRP3 release promotes oligomerization of NLRP3, ASC, and pro-caspase-1 into an active NLRP3 inflammasome. The inflammasome activates caspase-1, which in turn cleaves prointerleukin-1β (IL-1β) into its active form causing inflammation. Additionally, Gasdermin D (GSDMD) is cleaved by caspase-1 and forms a transmembrane pore leading to pyroptotic cell death.

apoptosis. This provides a potential explanation why tumors with amplification of Chr.22q11, harboring the *BID* locus, become highly susceptible to Aurora kinase inhibitors. Consistently, increasing BID expression in model cell lines increases sensitivity to these drugs (Bertran-Alamillo et al, 2023; Rizzotto et al, 2024).

These findings imply that MDM2 proteolysis may not be engaged for cell killing, but solely to promote cell cycle arrest. Intriguingly, however, Caspase-2 appears to proteolyze BID and MDM2 simultaneously, but apoptosis-susceptible cells die due to the rapid formation of truncated BID that directly targets BAX/ BAK at mitochondria (Rizzotto et al, 2024). Remarkably, removing BID from such cells reveals this duality and shows that MDM2 is processed in parallel by Caspase-2 to promote a p53 transcriptional response that may serve as a back-up mechanism when BID levels are low (Rizzotto et al, 2024). One relevant target is the BH3-only protein *PUMA*, needed to execute p53-induced cell death in response to DNA damage (Villunger et al, 2003). Yet, *PUMA* induction does not appear to be entirely p53-dependent but can also be transcriptionally induced by TNF in cells treated with Aurora kinase inhibitors (Sun et al, 2014).

Taken together, this suggests that PUMA and BID act in concert to kill susceptible cells with extra centrosomes and can substitute for each other. Hence, p53-deficient tumors may become strictly reliant on BID downstream of extra centrosomes for apoptosis induction and cancers that express BID variants with reduced cell-

death activity and may develop rapid drug resistance (Flores-Romero et al, 2022). In contrast, those 6% of solid cancers with high *BID* expression levels may qualify as good responders (Bertran-Alamillo et al, 2023). Additional studies are required to delineate all molecular details how cells decide to engage cell cycle arrest versus apoptosis. Upstream, however, engagement of PIDD1 by ANKRD26 at distal appendages of amplified centrosomes is critical for both outcomes (Rizzotto et al, 2024). Induction of apoptosis downstream of extra centrosomes may help explain findings in skin and liver cells where PLK4 overexpression fails to induce cancer when p53 is functional (Coelho et al, 2015; Kulukian et al, 2015). Similarly, we noted increased cell-death rates in premalignant progenitor B cells overexpressing MYC and PLK4 simultaneously that may explain lack of synergy during transformation (Braun et al, 2024).

## Beyond CIN—emerging roles of centrosome aberrations in cancer cells

CA-mediated multipolar spindle assembly, as initially proposed by Boveri, has long been considered as the underlying mechanism of CIN and tumorigenesis. Today it is well established that cultured cells undergoing multipolar divisions are unviable and die by apoptosis (Ganem et al, 2009; Silkworth et al, 2009; Weiss et al,

2022). To prevent multipolar anaphases and limit the detrimental effects of CA on cell survival, cells cluster their extra centrosomes to assemble a pseudo-bipolar spindle (Quintyne et al, 2005). However, centrosome clusters promote the formation of merotelic kinetochore-MT attachments, where a single kinetochore is connected to MTs emanating from both opposing spindle poles. Such incorrect attachments increase the frequency of lagging chromosomes, chromosome missegregation and DNA breaks, thereby providing a direct mechanistic link between CA and CIN (Ganem et al, 2009; Silkworth et al, 2009; Crasta et al, 2012). Whether the efficiency of centrosome clustering depends on the number of centrosomes present within a cell has not been examined systematically. However, human samples derived from high-grade breast, colon, and prostate tumors that contain highly amplified centrosomes form tight centrosome clusters, while the efficiency of clustering in cultured cancer cell lines seems to be lower and varies depending on the type of cancer (Pannu et al, 2014). Inefficient centrosome clustering may well lead to fitness defects or cell-death phenotypes observed in cells with CA, for example after experiencing delayed mitoses, by activating the mitotic surveillance pathway (reviewed by Phan and Holland, 2021). Importantly, however, a number of recent studies report CA-dependent processes that may well contribute to cancer but are seemingly not connected to their negative impact on mitotic fidelity (Fig. 3).

## Centrosomes in cancer cell motility

Similar to mitosis, amplified centrosomes seem to also affect interphase-associated processes in cancer cells. CA correlates with tumor progression showing moderate centrosome numbers (3–4 per cell) in low-grade CIN and excessive CA (>4 per cell) in many types of invasive carcinomas (Skyldberg et al, 2001; Sato et al, 2010; Chan, 2011; Marteil et al, 2018). This clinical interrelation between centrosome numbers and the degree of invasion suggests that amplified centrosomes might confer some advantage to cancer cells that go beyond spindle dynamics, affecting their migratory capacities

and subsequent metastatic outgrowth. The mechanisms how tumor cells disseminate from the primary tumor and spread into distant sites are still ill-defined in their detail. In 3D cell culture models, one proposed mechanism how CA promotes tumor cell invasion is increased recruitment of PCM proteins such as γ-tubulin and thus enhanced centrosomal MT nucleation (Godinho et al, 2014). In line with these findings, cancer cells with overly long centrioles form larger and over-active MTOCs due to increased centrosomal levels of pericentrin and γ-tubulin, which correlates with tumor aggressiveness and a worse prognosis (Marteil et al, 2018).

How can enhanced centrosomal MT nucleation promote cancer cell dissemination from the primary tumor? Directional cell migration consists of four consecutive steps: (1) protrusion of the leading edge, (2) adhesion to the extracellular matrix, (3) translocation of the cell body, and (4) retraction of the trailing edge (reviewed by Lauffenburger and Horwitz, 1996; Ridley et al, 2003). Actin polymerization generates the protrusive activity of the cell front termed lamellipodium, while actomyosin filaments generate contractile forces at the side and the cell's rear to propel the cell body forward. Upon reception of migration-stimulating signals, localized activation of the small GTPase RAC1 promotes actin polymerization and lamellipodium formation in the direction of the signal (Ridley et al, 1992). Importantly, MT growth to the leading edge activates RAC1 in migrating fibroblasts and thereby stimulates the formation of lamellipodial protrusions that are required for locomotion (Waterman-Storer et al, 1999). In this context, work from the Pellman lab has demonstrated that increased centrosomal MT nucleation triggers invasion via elevated activation of RAC1 in cancerous cells. Moreover, invasive protrusions were accompanied by degradation of components of the extracellular matrix, thus allowing collective migration of cancerous cells into the surrounding matrix (Godinho et al, 2014). The molecular details how dynamic MTs activate RAC1 to promote invasion remain unclear. However, these experiments clearly demonstrate that amplified centrosomes can trigger hyperactive MTOC activity and alterations in the MT cytoskeleton, which foster the ability of cancer cells to migrate (Fig. 3). In the future it

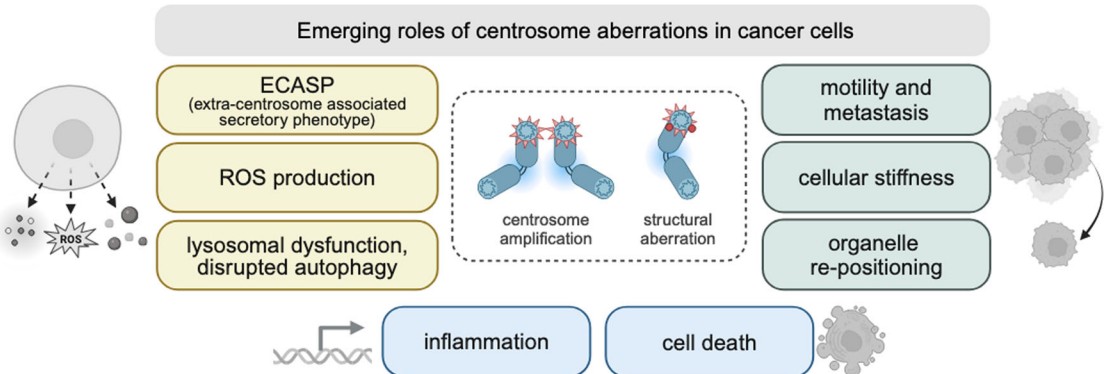

**Figure 3. Emerging roles of centrosome amplification in cancer cells.**

Aberrations in centrosome number and structure appear to have roles that go beyond multipolar spindle assembly and CIN in cancer cells. Cells with extra centrosomes have shown to cause altered chemokine and cytokine release (ECASP, extra centrosome-associated secretory pathway), ROS production, lysosomal dysfunction and disrupted autophagy (left yellow boxes). Furthermore, excessive CA can be observed in more invasive carcinomas and has been linked to motility, metastasis, cellular stiffness, and organelle re-positioning (right green boxes). Extra centrosomes can also activate NF-kB signaling and inflammation or BID-dependent apoptotic cell death in a PIDD1-dependent manner (lower blue boxes).

will be interesting to address whether and how centrosome configuration during interphase affects the migratory properties of normal as well as cancer cells and to elucidate whether centrosomal clustering is also a prerequisite for cancer cell motility and invasion.

## Centrosomes as modulators of cell stiffness

In addition to numerical aberrations, structural centrosome aberrations have been shown to affect cancer cell invasiveness via a non-cell-autonomous mechanism (Ganier et al, 2018b; Arnandis et al, 2018). The phenomenon was described in 3D epithelial cultures in which ninein-like protein (NLP) overexpression results in budding of mitotic cells towards the surrounding matrix. The underlying mechanism is driven by NLP overexpression interfering with E-cadherin junctions and altering cellular biomechanical properties. NLP overexpressing cells show enhanced MT stability and increased cellular stiffness, while cells lacking centrosome aberrations are "softer" and thus pushed out of the epithelia (Ganier et al, 2018a). This non-cell-autonomous form of centrosome alteration and the resulting increased invasiveness implies that metastatic properties may not be detectable in all tumor cells.

## Centrosomes as initiators of paracrine signaling

Modified centrosome-mediated cytokine secretion has recently been described in macrophages. Here, pathogen encounter, stimulated by LPS treatment, induces atypical centrosome maturation, which is accompanied by the recruitment of PCM components such as pericentrin, γ-tubulin, and ninein, causing increased MT-nucleation (Vertii et al, 2016). Interphase centrosome maturation depends on the mixed-lineage kinase (MLK) family but occurs independently of the mitotic kinase PLK1. Importantly, centriole depletion leads to attenuated secretion of a specific set of cytokines, namely IL-6, IL-10, and MCP1, while TNF production was unaffected. Together, this suggests a crucial role for the centrosome in regulating the release of specific cytokines upon inflammation by enhancing the secretory pathway. It remains to be explored if centrosome maturation in macrophages is also a prerequisite for pyrin or NLRP3 inflammasome recruitment, requiring HDAC6-dependent transport along MTs and/or inflammasome assembly at the centrosome, allowing for Caspase-1 activation and IL-1β processing (Magupalli et al, 2020).

Notably, the secretion of pro-invasive molecules, such as IL-8, drives invasion of human mammary epithelial cells with CA in 3D cultures, which is characterized by actin-rich protrusions and degradation of the basement membrane (Arnandis et al, 2018). Induction of paracrine invasion is mediated by oxidative stress and ROS production in cells with CA that leads to altered cytokine and chemokine secretion via the extra centrosome-associated secretory pathway (ECASP). This secretory phenotype resembles in part the SASP of senescent cells, hence a more detailed comparison with the ECASP seems warranted.

Recent work further emphasizes the impact of centrosome integrity on secretion, as amplified centrosomes were shown to compromise lysosomal function (Adams et al, 2021). The authors report that pancreatic cancer cells with amplified centrosomes produce elevated ROS levels leading to lysosomal dysfunction and the secretion of small extracellular vesicles. This enables communication with the surrounding stroma and subsequently promotes cell invasion. In line with these findings, PLK4-induced CA in RPE1 and MCF10A cells was reported to disrupt autophagosome trafficking and autophagy (Denu et al, 2020). Whether lysosome dysfunction is responsible for the observed deregulation of autophagy is currently not known. In addition to coordinating lysosome function, the centrosome acts as an intracellular organizer of various organelles which travel along MT filaments. Cells with extra centrosomes display changes in organelle positioning (Monteiro et al, 2023). Upon PLK4-driven CA, mitochondria, endosomes and intermediate filaments are reorganized and shifted to the cell periphery. Moreover, the centrosome is no longer located in close proximity to the nucleus. This global rearrangement of intracellular organelles is driven by MT-acetylation and enhances nuclear deformability, thus facilitating migration of cells with amplified centrosomes through locally confined microenvironments (Monteiro et al, 2023).

What remains unclear is how extra centrosomes actually lead to increased ROS levels and aberrant organelle positioning, including perturbations of the mitochondrial network (Adams et al, 2021; Monteiro et al, 2023). However, the increased production of cytokines and chemokines that facilitate invasiveness in cells that overexpress PLK4 might also be explained by recent observations that extra centrosomes can activate NF-kB signaling in a PIDD1-dependent manner. CA induced by PLK4 overexpression or induction of cytokinesis failure induces sterile inflammation in mouse embryonic fibroblasts and immortalized or transformed human model cell lines (Garcia-Carpio et al, 2023). This response helps to attract natural killer (NK) cells to eliminate cells at risk of developing more complex karyotypes (Garcia-Carpio et al, 2023). Elimination of cells with complex karyotypes that form in response to spindle assembly checkpoint inhibition using an MPS1 inhibitor (reversine) has been equally linked to sterile inflammation and NK-cell recruitment (Wang et al, 2021). As reversine also triggers cytokinesis defects in a good fraction of cells (Fava et al, 2017), it appears plausible that activation of the PIDDosome may contribute to the phenotypes noted in this and other studies that link aneuploidy and inflammation, independently of cGAS/STING signaling (Garcia-Carpio et al, 2023).

Together, these studies identify a number of centrosome-related functions, amplified or not, that in a cell-extrinsic or cell-autonomous manner can impact on cellular behavior, tumor progression, and cancer cell clearance. These observations may become relevant in the context of cancer treatment efficacy and side effects using newly developed inhibitors that interfere with centriole biogenesis or duplication, such as targeting PLK1, PLK4, or Aurora B kinase, or centrosome dynamics by interfering with proteins regulating centrosomal clustering, for instance by targeting HSET/KIFC1 (Kwon et al, 2008; Chavali et al, 2016; Vitre et al, 2020). Whether the same or similar processes noted here are critical for normal physiology and pathologies other than cancer is clearly understudied.

# Modulation of centrosome numbers as a function of normal physiology

Centriole structure and PCM composition are well known to be subject to alterations during organismal development and differentiation into specialized cell types. Even though centriole length is

generally stereotypic and tightly controlled, it may eventually differ between species and even in different cell types of one species (Jana et al, 2018). Such changes have been associated with differential regulation of centrosomal proteins. Thus, centrosome structure can likely be adapted to cell-type or tissue-specific needs (Carden et al, 2023). Similarly, centrosome numbers can also vary during development and differentiation, giving rise to cells that contain either no centrosomes or multiple thereof.

## Centriole loss during development and differentiation

Centriole elimination is observed in somatic cells of flies, *C. elegans* and some types of vertebrate and mammalian tissues (Fig. 4A), for example, during early embryogenesis or after whole-genome duplication (Mahowald et al, 1979; Bloomfield and Cimini, 2023). Cell differentiation often coincides with loss of centrosomal MT nucleation as observed in neurons, cardiac and skeletal muscle and keratinocytes (Stiess et al, 2010; Zebrowski et al, 2015; Bugnard et al, 2005; Muroyama et al, 2016) and reorganization of MT nucleation from noncentrosomal sites as described in the section "Centrosome structure". Mechanisms that instruct mammalian cells how many copies to produce in the absence of a template to avoid CA are still elusive (Xiao et al, 2021; Grzonka and Bazzi, 2024). Moreover, centriole loss is a hallmark feature of oocytes in various species to ensure appropriate zygotic centriole numbers when the sperm cell supplies centrioles during fertilization (Simerly et al, 1995). In *C. elegans*, terminal cell differentiation during embryogenesis is associated with the loss of centrioles while artificial maintenance of centrosomes impairs terminal differentiation (Kalbfuss and Gönczy, 2023). Here, centriole fate is stereotyped and the timing of elimination is characteristic of a given cell type. Curiously, loss of centrosomes has also been noted in cancer where a large heterogeneity in the number of centrioles

appears to co-exist within a single tumor (Morretton et al, 2022). Whether this correlated with the loss of "stemness" or show reduced proliferative capacity remains unclear though.

## Centrosome amplification during terminal differentiation

An increase in centrosome numbers has also been shown in certain differentiated cell types (Fig. 4B). In mice, hepatocytes acquire multiple centrosomes as a consequence of an impaired cytokinesis and polyploidization events at the time of weaning (Guidotti et al, 2003). About 40% of human hepatocytes are polyploid and carry extra centrosomes (Sladky et al, 2021). Of note, ploidy does not seem to affect liver function, nor regenerative proliferation in mice, as documented by comparing mononucleated diploid livers or highly polyploid livers of wild-type animals (Pandit et al, 2012; Zhang et al, 2018; Lin et al, 2020).

Until to date, the precise role of increased centrosome numbers for hepatocyte function has not been unveiled (Sladky et al, 2021). Moreover, how hepatocytes evade cell death or sterile inflammation in response to CA remains uncertain, but PIDD1 and Caspase-2 that limit liver ploidy and are required for both responses, are rapidly downregulated during organogenesis by inhibitory E2F family members E2F7 and E2F8 (Sladky et al, 2020). A similar phenomenon may control ploidy in the heart, as loss of these non-canonical E2F family members leads to reduced cardiomyocyte ploidy but has no impact on heart function or regeneration (Yu et al, 2023). Resembling findings made in the liver, also here the PIDDosome controls ploidy in postnatal cardiomyocytes (Leone et al, 2024, preprint). Remarkably, centriole cohesion in cardiomyocytes is lost shortly after birth leading to splitting of the two centrioles and loss of centrosome integrity during terminal differentiation (Zebrowski et al, 2015), probably as a measure to escape potentially unwanted side effects triggered by signaling events elicited via extra centrosomes discussed above.

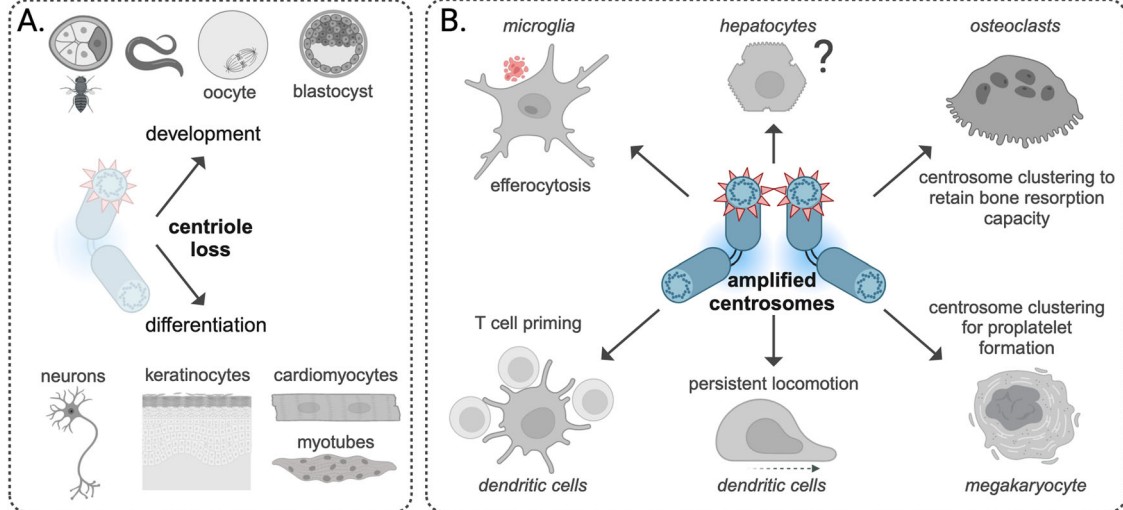

**Figure 4. Modulation of centrosome numbers as a feature of regular cell and tissue physiology.**

Variation of centrosome numbers is important during development and differentiation, giving rise to cells that contain either no centrosomes or multiple. Centriole elimination (A) is observed in somatic cells of flies, *C. elegans* and some types of vertebrate and mammalian tissues, for instance in early embryogenesis, neurons, keratinocytes, myotubes or after whole-genome duplication in cardiomyocytes. On the other hand, CA (B) has been shown to regulate T cell priming and persistent locomotion of dendritic cells to chemotactic cues, efferocytosis in microglia, bone resorption capacity in osteoclasts, proplatelet formation in megakaryocytes and has also been observed in polyploid hepatocytes.

Centriole splitting coincides with relocalization of PCM proteins such as pericentrin and CDK5RAP2 to the nuclear membrane. Both proteins are required for centriole cohesion (Graser et al, 2007; Matsuo et al, 2010).

In addition to hepatocytes and cardiomyocytes, centrioles in adult and embryonic mouse olfactory neurons can amplify from the progenitor cell via formation of centriole rosettes prior to cell division. The presence of amplified centrioles correlates with increased expression levels of PLK4 and STIL (Ching and Stearns, 2020). Whether these structures are fully mature and which function they fulfill, if any, remains to be defined. A similar finding has recently been made in proliferating progenitor B cells in the bone marrow of mice. A significant portion of highly proliferative pro- and pre-B lymphocytes were found to carry extra centrioles. Notably, these structures were no longer seen during later stages of B cell development nor in resting mature B cells (Schapfl et al, 2024). While the precise role of additional centrioles in early B cell development needs to be defined, it is evident that progenitor B cells can give rise to acute lymphatic leukemias. The possibility that these extra centrioles, even though they are part of a physiological program, may increase the risk of malignant transformation is an intriguing possibility.

Induction of CA by PLK4 overexpression in mature B cells even led to an enhanced capacity to process and present antigens (Yuseff et al, 2011). This implies that CA may be able to tune cellular immunity. Whether the immunological synapse still forms in the absence of centrioles has not been tested in B cells, but B cells devoid of centrioles due to ablation of *Plk4* can still mount a humoral immune response (Schapfl et al, 2024). Thus, it will be interesting to study how alterations in centriole number or centrosome function affect the quality of adaptive immunity.

Increased centrosome counts, however, clearly enhance effector function of antigen-presenting dendritic cells and microglia (Weier et al, 2022; Möller et al, 2022; Stötzel et al, 2024, preprint). Dendritic cells acquire extra centrosomes upon antigen encounter via mitotic defects and PLK2-mediated overduplication of centrioles, and nucleate a larger number of MT filaments compared to cells with only a single centrosome. Importantly, additional centrosomes do not compromise cellular fitness but instead promote directional migration of cells toward chemotactic cues and correlate with an increased capacity to activate CD4$^+$ T helper cells (Weier et al, 2022; Stötzel et al, 2024, preprint). These results demonstrate that the migratory capacity of immune cells can be modulated by extra centrosomes, highlighting that similar pathways and effector molecules may operate downstream of amplified centrosomes in cancer and immune cells.

A similar phenomenon of centrosome-mediated enhancement of immune-cell effector function was recently described in microglia, which are tissue-resident macrophages of the brain (Möller et al, 2022). The authors show that reorientation of the centrosome towards a forming phagosome is essential for successful branch-mediated efferocytosis and accompanied by the formation of a polarity axis via targeted endosome trafficking to the phagocytic synapse. Remarkably, artificial duplication of centrosomes enhanced the capacity of dead cell clearance, indicating that the centrosome plays a rate-limiting role in the process of neuronal efferocytosis.

Megakaryocytes, in charge of platelet production to control blood coagulation, increase cellular ploidy during maturation massively by endoreduplication, a process that also coincides with CA (Vitrat et al, 1998). These amplified centrosomes eventually form structures referred to as "supercentrosomes". Interference with clustering using HSET/KIFC1 inhibitors, or its genetic ablation in mice, leads to reduced proplatelet formation and platelet shedding in the vasculature, thereby linking cell cycle exit and centrosome clustering to cellular output (Becker et al, 2024).

Lastly, osteoclasts arise from macrophage progenitors that fuse during terminal differentiation and control bone resorption. Osteoclast function has been linked directly to cell size and ploidy (Bar-Shavit, 2007; Soysa et al, 2012). Moreover, osteoclasts arrange extra centrosomes into a defined cluster, which nucleates MT filaments, similar to the "supercentrosomes" reported in megakaryocytes. Impairing centrosomal clustering by either pharmacological inhibition or *NINEIN*-depletion markedly reduced bone resorption (Philip et al, 2022; Gilbert et al, 2024). Whether the size of "supercentrosomes" regulates the functional output, for instance by enhancing vesicle transport and the release of bone-resorbing enzymes, remains to be tested. If so, measures that reduce centrosome number during osteoclast differentiation, such as PLK4 inhibition, may delay bone-degenerative processes, including osteoporosis. Yet, in megakaryocytes, PLK4 inhibition may cause thrombocytopenia—an undesired side effect in the context of cancer treatment.

Together, these findings suggest that amplified centrosomes can be tolerated not only in proliferating cancerous cells but also in non-malignant cells as part of a defined developmental program, impacting on immunity to infection or adding to cellular output. We postulate that amplified centrosomes are not simple bystanders but instead are exploited by these cells to boost specific effector functions that are required upon changes of the extracellular microenvironment, such as during infection, stress-induced thrombocytopenia or ageing. However, how proliferating cells cope with additional centrosomes and whether centriole clustering is more efficient in these cells are open questions that await further investigation.

## Conclusions and perspectives

While the majority of our knowledge about CA events links amplified centrosomes to pathology, recent work suggests that extra centrosomes can also control terminal differentiation, best established in hematopoietic cells, in order to improve effector function. Whether extra centrosomes may assist hepatocyte specific functions in certain contexts remains to be defined. However, since these structures are maintained and not inactivated or actively lost, as seen in the mammalian heart, or in *C. elegans*, respectively, it is fair to assume that their maintenance is biologically significant, for example, to improve vesicle transport and secretion or migration in response to liver damage.

Moreover, centrosomes appear to be able to ignite signaling events that are unique and differ from those elicited in cells with normal centrosome count, such as orchestration of DNA repair or mitotic entry, as well as cytokine secretion or the nucleation of pyroptosis signaling in response to pathogens. Amplified centrosomes can promote sterile inflammation, enhance migration and add to invasiveness—maybe not just in cancer cells—but can also trigger apoptotic cell death. This apoptotic response can be

executed independent of p53, which may be harnessed for cancer therapy. As such, the efficacy of mitotic drivers, such as MPS1 or CENP-E inhibitors but also Aurora kinase inhibitors that interfere with cytokinesis, will clearly depend in part on the ability of amplified centrosomes to elicit mitochondrial apoptosis. Yet, the activity of these drugs may not solely rely on apoptosis, but also by enforcing tumor immune recognition by NK-cell attraction.

Given all of the above, it remains unclear how cancer cells with amplified centrosomes evade apoptosis. The ability to cluster them for the formation of pseudo-bipolar spindles (Quintyne et al, 2005) and frequent loss of p53 in such cells can only be a part of the equation. De-clustering agents show promising in vitro efficacy (Pannu et al, 2014; Raab et al, 2012) but how they kill cancer cells and whether they affect non-malignant cells that cluster extra centrosomes, such as megakaryocytes or osteoclasts, remains to be explored further.

Taken together, accumulating evidence supports a view that amplified centrosomes cannot be regarded as an ever-consistent danger to cellular and organismal health. We should rather look at these structures from different angles, as friend or foe of tissue homeostasis, as desired feature of cellular differentiation processes, or maybe just as a glitch in the system, when orchestration of multiple signaling events that coordinate proliferation with differentiation is difficult to achieve.

# Peer review information

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

## Acknowledgements

We thank Oliver Gruß for critically reading the manuscript and helpful discussion. Figures were created with BioRender. The research of EK is supported by a fellowship of the Ministry of Innovation, Science and Research of North-Rhine-Westphalia (AZ: 421-8.03.03.02-137069) and the Deutsche Forschungsgemeinschaft (DFG, German Research Foundation) under Germany's Excellence Strategy – EXC 2151 – 390873048. This research of AV is funded in part by the Austrian Science Fund (FWF) [10.55776/DOC82], [10.55776/I6642], [10.55776/P36658] and the European Research Council, ERC AdG POLICE (787171). For open access purposes, the authors have applied a CC BY public copyright license to any author accepted manuscript version arising from this submission.

## Author contributions

**Eva Kiermaier**: Conceptualization; Supervision; Funding acquisition; Writing—original draft; Writing—review and editing. **Isabel Stötzel**: Visualization; Writing—original draft; Writing—review and editing. **Marina A Schapfl**: Visualization; Writing—original draft; Writing—review and editing. **Andreas Villunger**: Conceptualization; Supervision; Funding acquisition; Writing—original draft; Writing—review and editing.

## Funding

## Disclosure and competing interests statement

The authors declare no competing interests.

