## [Peer Review File · EMBO Reports]

Amplified centrosomes - more than just a threat

Eva Kiermaier, Isabel Stötzel, Marina Schapfl, and Andreas Villunger

Corresponding author(s): Eva Kiermaier (eva.kiermaier@uni-bonn.de) , Andreas Villunger (andreas.villunger@i-med.ac.at)

Review Timeline:

Submission Date:	19th Apr 24
Editorial Decision:	20th Jun 24
Revision Received:	5th Jul 24
Accepted:	28th Aug 24

Editor: Deniz Senyilmaz Tiebe

Transaction Report:

Dear Eva,
Dear Andreas,

Thank you for the submission of your review article to our editorial offices. I have received the full set of referee reports that is copied below, which I had shared with you earlier. As you will see, all three referees state that your manuscript is interesting and timely. However, they have several suggestions to improve the submission that I kindly ask you to address in a revised manuscript.

Given the constructive referee comments, I would thus like to invite you to revise your manuscript with the understanding that all referee points are addressed in the revised manuscript and in a detailed point-by-point response.

I further have these editorial requests:

- Please provide the manuscript text in the word format.
- We note that there are currently six keywords. However, the maximum number of keywords we can accommodate is five. Thus, please remove one of the keywords.
- Please move the Figure Legends to the end of the manuscript.
- We note that the individual panels of the figures (A-B) are currently not called out in the text.
- Please add a short section entitled "Box 1: In need of answers", that allows you to describe open questions in the field in the form of a bullet pointed list. Please refer to it in the text by including a callout.
- Please remove the ORCIDs from the manuscript title page.
- Please remove the Author Contributions section from the manuscript.
- Please remove the information regarding BioRender from the figure legends. Instead, please refer to it in the Acknowledgements section.

I think this is a very interesting review and while I appreciate that incorporating the referees' suggestions will still require some work, I am convinced that the article is worth it and will benefit from it.

When submitting your revised manuscript, we will require a Microsoft Word file (.doc) of the revised manuscript text including detailed figure legends (at the very end), but without the figures.

Please provide the final figures as separate, high resolution files as .pdf, .eps, .tif, or .jpg (one file per figure). Please finalize the drafts provided and make sure they accurately illustrate the key scientific concepts that you wish to show.

Please also note the following points:

- If there are certain aspects of your figure draft that are based upon assumptions or where the scientific data remains ambiguous (for example, schematically depicting a presumed direct protein-protein interaction, protein shape or subcellular localizations etc.) please add a comment so that we can work with you on an accurate depiction. Please ensure the directionality and nature of interactions is presented accurately.
- If the figure or single panels of the figure have been adapted from a published figure, please add this information to the figure legend (e.g., 'Adapted from...' or 'Based on...'). The editor will discuss if a reference and permission will be necessary
- Please only re-use figures or parts of a figure if this is essential for understanding the concept communicated. Often a reference to a previous paper will suffice. If the figure contains re-used images or elements of images, including schematics, micrographs or photos, please make sure that you have the permission/license to publish it (this also applies to your own previous work, if the journal you published in retains copyright. Certain 'creative commons' open access licenses, such as CC-BY 4.0, allow re-use without additional formal permissions). All re-used material must be explicitly cited.
- If you use an image data base for scientific iconography (e.g., BioRender), please let us know if you have a license that allows for publication in an academic journal. Often authors use misleading iconography for expedience. Please ensure the information shown is scientifically accurate. If in doubt, please discuss with the editor or provide a sketch so that our designers can create accurate iconography.
- For figures created using a software for editing vector objects like Inkscape, CorelDraw etc., please send the file as a PDF (or SVG, or EPS), PowerPoint or Keynote in which the labels and objects are still editable. For figures created using Adobe Illustrator, please send the Illustrator (.ai) file.

I look forward to seeing a revised version of your manuscript when it is ready. Please let me know if you have questions or comments regarding the revision.

Kind regards,

Deniz

Deniz Senyilmaz Tiebe, PhD
Scientific Editor
EMBO Reports

Referee #1:

The review by Kiermaier and colleagues on "Amplified centrosomes- more than just a threat" is a very interesting perspective on a less discussed side of the role and consequences of centrosome amplification. A potential beneficial role for centrioles and their amplification in signalling, is a field that only recently started to be explored and a review was much in need.

Overall, the text is well balanced, and the figures are very nice. I just have a few comments:

1-The authors just speak about the centriole-containing centrosome. In fact, there is evidence associating yeast centrosomes (SPBs) to signalling. It would be nice to also refer to the yeast centrosome, while saying that the focus will be on the animal counterpart (reviewed in Langlois-Lemay and D'Amours, 2022).

2-lines 100-102, when speaking about γ -tubulin independent nucleation processes it would be interesting to refer to which molecules are responsible for that.

3-line 136, a very important aspect on controlling the number of procentrioles is PLK4 degradation, triggered by PLK4 autophosphorylation (labs of Rogers, Bettencourt-Dias, Holland and Nigg), which the authors do not refer to.

4-line 146, cyclinB/CDK1 also prevents further reduplication by competing with PLK4 for STIL binding (Zitouni, Current Biology, 2016)

5-line 175- this part (part 4) while very interesting is very long and a bit confusing. Can the authors revise it and restructure it, perhaps with some subtitles?

6-line 208- any reason why amplification of that region makes them more susceptible

7-line 278- Arandis et al Dev Cell (2018) also refers to non-cell autonomous mechanisms.

8-line 336 Not sure about the term centrosome dissolution. Perhaps attenuation of centrosome function? Centrosome inactivation (it is there but inactive)? Or centrosome elimination (is not there anymore), depending on what the authors are referring to.

9-In figure 1B-red- the symbol for migration is not clear for me.

10-In figure 3B, it is not clear (for non-centrosome people) that it refers to centrosome amplification- should be more explicit.

Referee #2:

The review by Kiermaier et al. discusses how an increase in the number of centrosomes, which is normally tightly regulated, is linked to various pathological conditions including cancer. The authors then make the point that not all alterations in centrosome number are necessarily pathological, but have in fact important roles in normal development and physiology. Overall this manuscript is timely and addresses an important topic. While I do not see any major issues, I feel that there is significant lack of coherence between text and figures and that various aspects need revision.

Issues:

1) General: I would strongly advise to use more precise language (see some examples in the following points). Also, what seems to be the main topic of the review, the role of extra centrosomes in normal physiology, is actually discussed only in the last section. I felt that this part was too short and superficial, considering the expectations that are raised by title and abstract. It would also be useful to include more systematically (perhaps at the end of sections) conclusions, interpretations, or hypotheses based on the available data, rather than simply summarizing the literature.

2) General: there is no discussion of the importance of centrioles for generating cilia. Work from the Stearns lab has shown impaired primary cilium signaling when cells with extra centrioles assemble multiple primary cilia. This could in fact contribute to phenotypes in cells with less or extra centrosomes, in particular during development and tissue homeostasis.

3) General: there is no discussion of non-centrosomal MTOCs, which are crucial in cells that lose their centrosomes.

4) There is no discussion of the complete loss of centrioles as a point-of-no-return. For example, if a cell were to re-enter the cell cycle, without pre-existing centrioles, there is not restriction of centriole number and de novo centrioles would form with abnormal numbers.

5) Terms and definitions: the authors use various terms to describe the same structure or process, including terms that are not

commonly used (I don't think that "multi-numerous" is an existing term. It sounds intrinsically redundant and I would advise against it). I suggest to define these terms carefully and then stick with a specific terminology, to ensure that the text is clear and precise. For example, the authors talk about extra, multiple, amplified, and multi-numerous centrosomes, but also about numerical aberrations, abnormalities, multiplication, impaired integrity etc. Do all of these refer to the same condition? What exactly is the definition?

6) Don't the authors think that it would make a difference to have one extra centrosome e.g. due to failed cytokinesis vs. having 4, 5, or 6...? This should be addressed by defining the condition of having extra centrosomes clearly in the beginning (see above) and by discussing potential differences in the consequences.

7) p2: I think the statement that amplified centrosomes have been identified in virtually all types of human cancers is too strong, at least the provided references do not show this. In my opinion the same applies to the statement that it is well established that amplified centrosomes are sufficient to induce transformation and cancer. In the absence of pre-existing p53 KO, this was observed only in a single study in mice (Levine et al 2017), using specific experimental conditions (PLK4 overexpression) and affecting only specific tissues. The details and limitations of findings should also be mentioned and discussed, since they are important.

8) Section 2: First, the authors state that PCM lacks a "rigid" ultrastructure. What does "rigid" refer to? I am not aware of any study that investigates rigidity of the PCM (structured does not necessarily mean rigid). Then, in the following sentences the authors state that the PCM forms "defined concentric layers" - isn't this a structured organization and contrasts with the previous statement? In my opinion, the super resolution studies have debunked the idea of the PCM being amorphous (at least in vertebrate interphase cells). I would also not refer to these studies as "recent", since they were published 12 years ago. Some references are also missing here (Agard, Nigg, Glover,...).

9) Throughout text: gamma-tubulin/gammaTuRC is spelled with a "y", not the Greek gamma.

10) Is the phase separation idea for PCM proteins relevant here? I would remove it or describe it more nuanced, since it is not generally accepted, at least not in vivo and for organisms other than *C. elegans*. The cited Raff 2019 reference is actually a very critical discussion of this concept and not a support, as suggested by the context in which it is cited.

11) The discussion of gamma-tubulin-independent nucleation: again, I wonder if this is relevant here? If so, it should also mention previous work by the Goshima lab.

12) Overall, the duplication cycle is well and correctly described (unfortunately not always the case even in expert reviews), just a couple of minor points:

Line 106: the authors seem to use "duplication" and "multiplication" interchangeably. They are not the same, and "multiplication" is generally not used for normal centriole and DNA duplication/replication. In fact, in other places of the manuscript multiplication is used to refer to extra centrosomes.

Line 110: Centrioles are not tethered at their distal ends, but at their proximal ends.

13) Line 150: "Impaired centriole duplication can lead to overduplication" - more precise language is needed here: how can impairment lead to more duplication rather than no duplication? I assume the authors refer to impaired regulation, premature disengagement, etc, but this should be stated.

14) I found the text sections starting on page 6 very hard to follow. Several statements are based on assumptions that are not obvious to the non-expert reader. Example:

"The pro-apoptotic BCL2 family protein BID serves as a Caspase-2 substrate, becomes activated and promotes the rapid apoptosis induction in human blood cancer cell lines (Rizzotto et al, 2024). This finding explains why inhibition of Aurora kinase, leading to cytokinesis failure, can also kill p53-deficient cancer cells (Sun et al, 2014) and why tumors/cells with amplification of Chr. 22q11 are highly susceptible to this treatment strategy (Bertran-Alamillo et al, 2023)."

I cannot follow the line of thoughts presented here without a better, more detailed explanation, perhaps also split up into several sentences. This is just one example.

15) Section 5 states "Beyond CIN", but CIN has not been introduced before. In fact, there is not discussion of why cells should control centrosome numbers and maintain a single centrosome in cycling cells. How is this linked to their roles in mitosis and assembly of primary cilia?

16) The same section heading states "emerging roles of extra centrosomes". To me this sounds as if physiological functions are going to be discussed, but the section only deals with disease conditions. On page 8 the discussion switches to impaired centrosome structural integrity, which does not fit with the above heading. Similarly, the following section 6 heading states to discuss extra centrosomes in normal physiology, but begins with a discussion of abnormal centriole structure and PCM composition. Then, rather than extra centrosomes, loss of centrosomes is discussed.

17) The authors state several times that centrosomes "dissolve" in cardiomyocytes, please revise this term.

18) Sometimes the authors talk about extra centrioles and sometimes about extra centrosomes (e.g. lines 344 and 346, but also elsewhere). This is clearly not the same thing, but what is the difference and why does it matter? These issues need more precise language and explanation for non-expert readers.

19) Figures: Overall, I found the figures to be quite disconnected from the text. Some parts are not discussed at all.

The structure of Figure 1 is not replicated in the text, e.g. there is no real discussion of centriole loss and structural defects and the related pathologies ciliopathy and microcephaly. Also, the cartoons used here display normal structures rather than diseased conditions (compared to the cartoons on the right). Similarly, under "physiology" some terms on the left simply name cell types rather than physiological processes. It is unclear what exactly is shown here.

Figure 2 displays signaling "downstream of extra centrosomes", but an extra centrosome is shown only in A. Is this on purpose? Does this apply only to having one extra centrosome as depicted in the cartoons? Does the number matter? I have not found any discussion of panel B in the text.

Figure 3 shows non-canonical functions of centrosomes. Again, I did not find a discussion of this in the text. This figure is cited in the context of extra centrosomes. "Increased PCM and microtubules" in B refers to an increased canonical function, but is not a non-canonical function.

Referee #3:

This review article discusses the consequences of extra centrosomes in pathological and physiological conditions, highlighting how these extra centrosomes are not just deleterious, but can be part of physiological process.

Overall, the review is very pleasant to read, presents an in depth and balanced discussion on the topic and its subject is timely. It does not just present a list of findings, but highlights a nice alternative to the prevalent hypothesis that extra centrosomes are mostly a dangerous and a pathological condition.

I have a number of minor concerns/comments that are listed below. The authors should consider those as suggestions that could further improve this review article:

- On page 2 and page 7 the authors allude to the role of extra centrosomes in generating chromosomal instability, but never present the mechanistic detail. Stating that extra centrosomes favor the formation of transient multipolar spindles that increase the proportion of erroneous kinetochore microtubule attachments that persist as cells enter anaphase (Ganem et al., 2009 but also Silkworth et al., 2009 that appeared at same time), would be helpful.
- bottom of page 3: The authors might consider discussing briefly other structures contributing to microtubule nucleation in the mitotic spindle, such as the augmin pathway (work of the Gerlich laboratory) and kinetochore-driven microtubule nucleation (Maiato et al., 2004 and the recent work of the Kops laboratory showing that this MT nucleation at kinetochores depends also on pericentrin)
- page 4: When discussing the dissolution of the linker between the two centrosomes the author could mention the key role of the Nek kinases (work of the Fry and Roig laboratories)
- page 5: when discussing the coordination of DNA replication and centrosome duplication at the G1/S boundary, authors could cite the initial studies linking centrosome duplication to Cdk2 activity (Hinchcliffe et al., 1999; Meraldi et al., 1999 (I apologize for self-citing); Lacey et al., 1999 and Matsumoto et al., 1999)
- page 5: when presenting mechanisms leading to extra centrosomes, the authors might want to also include the concept of breakage of excessively long centrioles, first proposed by the Bettencourt laboratory (Marteil et al., 2018)
- page 5: when discussing how chromosome segregation errors and an abrogated mitosis can lead to extra centrosomes the authors might also include the original study describing such events (Meraldi et al., 2002)
- page 6: when talking about the mitotic surveillance pathway the authors might include a review article on the topic such as Phan and Holland 2021.
- page 7: when discussing extra centrosomes correlating with cancer grades the authors should again cite Marteil et al. 2018

- on page 10: when discussing the unknown role of extra centrosomes in the liver the authors might want to present the work of Duncan et al., Nature, 2010, which indicated that extra centrosomes can lead to multipolar spindles that favor a reduction in ploidy.

Patrick Meraldi

Response to the Reviewers' comments on EMBOR 2024-59450V1 'Amplified centrosomes – more than just a threat'

We thank all three reviewers for the time and effort dedicated to review our manuscript and their insightful feedback on our work. We believe that this constructive criticism, which we have now addressed, greatly improved our revised manuscript. Please find below our point-by-point response to the specific comments and suggestions.

Referee #1:

The review by Kiermaier and colleagues on "Amplified centrosomes- more than just a threat" is a very interesting perspective on a less discussed side of the role and consequences of centrosome amplification. A potential beneficial role for centrioles and their amplification in signalling, is a field that only recently started to be explored and a review was much in need.

Overall, the text is well balanced, and the figures are very nice. I just have a few comments:

1-The authors just speak about the centriole-containing centrosome. In fact, there is evidence associating yeast centrosomes (SPBs) to signalling. It would be nice to also refer to the yeast centrosome, while saying that the focus will be on the animal counterpart (reviewed in Langlois-Lemay and D'Amours, 2022).

We thank the reviewer for pointing out yeast spindle pole bodies as functional analogue of the centrosome – not only in terms of MT organization but also regarding the formation of signaling hubs. In the revised version, we now briefly mention the fungi counterpart and refer the reader to the suggested review at the end of the introductory paragraph for more detailed insights on SPBs and downstream signaling (page 3; line 95).

2-lines 100-102, when speaking about γ -tubulin independent nucleation processes it would be interesting to refer to which molecules are responsible for that.

We included a sentence on the protein Mini-spindles (MSPS) on page 4; line 115, which promotes MT nucleation in the absence of γ -TuRC (Zhu et al., JCB, 2023).

3-line 136, a very important aspect on controlling the number of procentrioles is PLK4 degradation, triggered by PLK4 autophosphorylation (labs of Rogers, Bettencourt-Dias, Holland and Nigg), which the authors do not refer to.

We thank the reviewer for raising the important point of PLK4 autophosphorylation and degradation. We included a paragraph on this topic when we discuss the regulation of centrosome numbers on page 5; line 162 in the revised version of the manuscript.

4-line 146, cyclinB/CDK1 also prevents further reduplication by competing with PLK4 for STIL binding (Zitouni, Current Biology, 2016)

We comment on CDK1 function in preventing untimely PLK4-STIL assembly to regulate centriole biogenesis and added the suggested reference (page 6; line 180) in the revised version of the manuscript.

5-line 175- this part (part 4) while very interesting is very long and a bit confusing. Can the authors revise it and restructure it, perhaps with some subtitles?

We now included subheadings in section 4 to structure this part better and to make it more concise. We hope this will help to break down a wealth of information into more digestive pieces. In addition, we included a short paragraph about the inflammasome and refer to Figure 2B to illustrate better the analogies between the PIDDosome complex and the inflammasome, both of which can induce cell death in a centrosome-dependent manner (page 7-8).

6-line 208- any reason why amplification of that region makes them more susceptible

This information may have been lost, but the amplified genomic region harbors the BID locus that leads to higher expression levels of this BH3-only protein and hence increased cell death in response to Aurora kinase inhibition, which triggers CA by cytokinesis failure. We hope this message comes now better across (page 8; line 266).

7-line 278- Arnandis et al Dev Cell (2018) also refers to non-cell autonomous mechanisms. *We included the suggested reference when we refer to non-cell autonomous mechanisms (page 11; line 350).*

8-line 336 Not sure about the term centrosome dissolution. Perhaps attenuation of centrosome function? Centrosome inactivation (it is there but inactive)? Or centrosome elimination (is not there anymore), depending on what the authors are referring to.

We apologize for this confusion – as pointed out also by reviewer 2. We replaced the term ‘centrosome dissolution’ by ‘loss of centriole cohesion’ as the work of Zebrowski et al. demonstrates that centrioles split shortly after birth leading to loss of centrosome integrity. Centriole splitting correlates with relocalization of pericentrin and CDK5RAP2 to the nuclear membrane that then takes over MTOC function (page 13; line 454).

9-In figure 1B-red- the symbol for migration is not clear for me.

The picture should show a transmigrating T cell, which is probably not the best example for migration but rather represents cell squeezing through narrow pores. We revised this Figure and now show in Figure 4 different cell types that contain amplified centrosomes and the function that we believe is influenced by amplified centrosomes. The migrating cell is now represented by an amoeboid migrating leukocyte in Figure 4.

10-In figure 3B, it is not clear (for non-centrosome people) that it refers to centrosome amplification- should be more explicit.

We also revised Figure 3 (now Figure 4) and labelled amplified centrosomes accordingly as well as the associated processes that we believe are impacted by CA.

Referee #2:

The review by Kiermaier et al. discusses how an increase in the number of centrosomes, which is normally tightly regulated, is linked to various pathological conditions including cancer. The authors then make the point that not all alterations in centrosome number are necessarily pathological, but have in fact important roles in normal development and physiology. Overall this manuscript is timely and addresses an important topic. While I do not see any major issues, I feel that there is significant lack of coherence between text and figures and that various aspects need revision.

We are pleased to hear that this referee considers our review as timely and that they have taken the time to help to improve content and coherence.

Issues:

1) General: I would strongly advise to use more precise language (see some examples in the following points). Also, what seems to be the main topic of the review, the role of extra centrosomes in normal physiology, is actually discussed only in the last section. I felt that this part was too short and superficial, considering the expectations that are raised by title and abstract. It would also be useful to include more systematically (perhaps at the end of sections) conclusions, interpretations, or hypotheses based on the available data, rather than simply summarizing the literature.

We have carefully considered the points raised, which, in our opinion, greatly helped to improve our manuscript. We agree that the emerging roles of extra centrioles may appear

somewhat “under-weight” within this review in light of the chosen title, but it was simply impossible for us to cut the wealth of knowledge related to centrosome biogenesis and cancer even shorter. Together with the thoughtful feedback provided by the other two reviewers, and addressing the suggestions made by this referee, we fear this may not have changed weights in the revised version.

However, we have made a serious effort to use more precise language to make it clear to experts and also non-experts what we aim at referring to (see also point 5). Moreover, we include now more thorough interpretations based on the cited literature and aim to formulate clear hypotheses on the role of amplified centrosomes on somatic cells. We hope that with these modifications, we could satisfy the reviewer’s concerns.

2) General: there is no discussion of the importance of centrioles for generating cilia. Work from the Stearns lab has shown impaired primary cilium signaling when cells with extra centrioles assemble multiple primary cilia. This could in fact contribute to phenotypes in cells with less or extra centrosomes, in particular during development and tissue homeostasis.

We agree with the reviewer that CA in the context of primary cilium assembly and signaling is of great importance and we failed to discuss this in our review. However, this was stated in the introduction. To alleviate this shortcoming to some degree, we now mention CA in polycystic kidney and liver diseases (**page 3; line 76**) and refer the reader to an excellent recent review from the Holland lab, that discusses the consequences of centriole amplification in the context of multi-ciliation and associated pathologies (**page 3; line 94**).

3) General: there is no discussion of non-centrosomal MTOCs, which are crucial in cells that lose their centrosomes.

We thank the reviewer for that comment which was also pointed out by reviewer 3. We added a paragraph about non-centrosomal MT nucleation **on page 4; line 118**, in which we discuss alternative structures, that have been shown to act as MT organizing sites in cells and included work from the Goshima and Gerlich lab about the Augmin complex and Augmin accumulation on long-lived MTs, the Khodjakov lab on kinetochore-driven MT nucleation and a recent review from Sabine Petry on branching MT nucleation.

4) There is no discussion of the complete loss of centrioles as a point-of-no-return. For example, if a cell were to re-enter the cell cycle, without pre-existing centrioles, there is not restriction of centriole number and de novo centrioles would form with abnormal numbers.

We acknowledge a certain gap in our review related to this topic. Yet, we have discussed the mitotic surveillance pathway in the context of centrosome loss, as well as the phenomenon on centriole depletion on **pages 7; line 221**, respectively. We now also include the observation that cell divisions in the early mouse embryo happen in the absence of extra centrosomes and that it is unclear how CA is avoided if de novo biogenesis of centrioles occurs without a template (**page 13; line 439**). In case this referee has a set of additional publications in mind, we can certainly expand the discussion on the topic.

5) Terms and definitions: the authors use various terms to describe the same structure or process, including terms that are not commonly used (I don't think that "multi-numerous" is an existing term. It sounds intrinsically redundant and I would advise against it). I suggest to define these terms carefully and then stick with a specific terminology, to ensure that the text is clear and precise. For example, the authors talk about extra, multiple, amplified, and multi-numerous centrosomes, but also about numerical aberrations, abnormalities, multiplication, impaired integrity etc. Do all of these refer to the same condition? What exactly is the definition?

We absolutely agree with the reviewer that it is helpful for the reader to stick with a certain terminology in particular when we talk about ‘amplified’ centrosomes. We define CA as state with 2 or >2 centrosomes in G1 phase of the cell cycle (we now included this information in the abstract). Most of the time, CA is considered as >2 centrosomes but we feel that with this definition we neglect differentiated cells which contain ‘only’ 2 centrosomes in G1 phase. Due

to the fact that we would expect only one centrosome in G1, we include this also under the term 'amplified centrosomes'.

Besides 'amplified centrosomes', we also use the term 'extra' centrosomes as it is frequently used by colleagues to describe more than the regular number of centrosomes (e.g. extra-centrosome-associated secretory pathway; introduced by Arnandis et al., 2018). We completely deleted the terms 'multi-numerous' and 'multiple' from our manuscript. Similarly, we made sure that we use only the term 'centrosome aberration', when we talk about numerical and structural centrosome alterations and deleted the terms 'abnormalities' and 'impaired integrity'.

6) Don't the authors think that it would make a difference to have one extra centrosome e.g. due to failed cytokinesis vs. having 4, 5, or 6,...? This should be addressed by defining the condition of having extra centrosomes clearly in the beginning (see above) and by discussing potential differences in the consequences.

*We thank the reviewer for this comment. We do indeed believe that the number of extra centrosomes matters, in particular when it comes about organizing these additional centrosomes during mitosis, but also in interphase. We added a paragraph on the topic 'centrosome clustering' and whether the number of centrosomes matters in this process on **page 9, line 304** in the revised version of the manuscript.*

7) p2: I think the statement that amplified centrosomes have been identified in virtually all types of human cancers is too strong, at least the provided references do not show this.

*We thank the reviewer for that comment and refined our list of cancer types that have been associated with centrosome amplification and added the respective references. Moreover, we replaced the term 'virtually all types of human cancers' by 'various human cancers' on **page 2; line 53**.*

In my opinion the same applies to the statement that it is well established that amplified centrosomes are sufficient to induce transformation and cancer. In the absence of pre-existing p53 KO, this was observed only in a single study in mice (Levine et al 2017), using specific experimental conditions (PLK4 overexpression) and affecting only specific tissues. The details and limitations of findings should also be mentioned and discussed, since they are important.

*We absolutely agree with the reviewer that it is still a matter of debate whether CA is one of the underlying causes of tumorigenesis as the experimental systems that have been used to prove causalities, in particular in mammals, rely solely on the use of PLK4 overexpression to induce CA, or only cause tumors in the absence of functional p53. We further elaborated and commented on these limitations in the revised version of the manuscript on **page 3, line 66**. We also discuss why CA might fail to drive disease in some tissues, such as the liver or skin on **page 9; line 288**.*

8) Section 2: First, the authors state that PCM lacks a "rigid" ultrastructure. What does "rigid" refer to? I am not aware of any study that investigates rigidity of the PCM (structured does not necessarily mean rigid). Then, in the following sentences the authors state that the PCM forms "defined concentric layers" - isn't this a structured organization and contrasts with the previous statement? In my opinion, the super resolution studies have debunked the idea of the PCM being amorphous (at least in vertebrate interphase cells). I would also not refer to these studies as "recent", since they were published 12 years ago. Some references are also missing here (Agard, Nigg, Glover,...).

*We apologize for not being precise here. In fact, we sought to describe how the PCM used to be characterized – as lacking a defined ultrastructure. Due to advances in superresolution techniques, the concept of concentric layers has been introduced demonstrating that also the PCM shows a certain degree of higher order structure, as mentioned by this referee. We rephrased the section to make this more clear and added the missing references (**page 4, line 101**).*

9) Throughout text: gamma-tubulin/gammaTuRC is spelled with a "y", not the Greek gamma.
We changed 'y-tubulin' to the greek gamma.

10) Is the phase separation idea for PCM proteins relevant here? I would remove it or describe it more nuanced, since it is not generally accepted, at least not in vivo and for organisms other than *C. elegans*. The cited Raff 2019 reference is actually a very critical discussion of this concept and not a support, as suggested by the context in which it is cited.

We agree with the reviewer that the concept of phase separation for PCM proteins is beyond the scope of this review and removed the respective section in the revised version of the manuscript.

11) The discussion of gamma-tubulin-independent nucleation: again, I wonder if this is relevant here? If so, it should also mention previous work by the Goshima lab.

We feel that this very recent work deserves attention as it has been believed for decades that all MTOCs rely on γ -tubulin. In this study, the authors demonstrate, that when centrosomes fail to recruit γ -tubulin complexes, they still nucleate microtubules via the protein Mini-spindles (MSPS). As also pointed out by reviewer 1, we now added the information about MSPS in the revised version of the manuscript (page 4; line 115).

12) Overall, the duplication cycle is well and correctly described (unfortunately not always the case even in expert reviews), just a couple of minor points:

Line 106: the authors seem to use "duplication" and "multiplication" interchangeably. They are not the same, and "multiplication" is generally not used for normal centriole and DNA duplication/replication. In fact, in other places of the manuscript multiplication is used to refer to extra centrosomes.

We appreciate that the reviewer acknowledges our summary of the centrosome duplication cycle. We exchanged 'multiplication' by 'duplication' on page 5; line 133 (former line 106).

Line 110: Centrioles are not tethered at their distal ends, but at their proximal ends.

We apologize for that mistake and corrected it.

13) Line 150: "Impaired centriole duplication can lead to overduplication" - more precise language is needed here: how can impairment lead to more duplication rather than no duplication? I assume the authors refer to impaired regulation, premature disengagement, etc, but this should be stated.

We thank the reviewer for this comment. We rephrased 'Impaired centriole duplication can lead to overduplication' to 'Impaired regulation of the centriole duplication machinery' (page 6, line 186). As an example we further explain how altered expression levels of proteins such as PLK4 lead to excessive centriole growth and overduplication of centrioles.

14) I found the text sections starting on page 6 very hard to follow. Several statements are based on assumptions that are not obvious to the non-expert reader. Example: "The pro-apoptotic BCL2 family protein BID serves as a Caspase-2 substrate, becomes activated and promotes the rapid apoptosis induction in human blood cancer cell lines (Rizzotto et al, 2024). This finding explains why inhibition of Aurora kinase, leading to cytokinesis failure, can also kill p53-deficient cancer cells (Sun et al, 2014) and why tumors/cells with amplification of Chr. 22q11 are highly susceptible to this treatment strategy (Bertran-Alamillo et al, 2023)."

I cannot follow the line of thoughts presented here without a better, more detailed explanation, perhaps also split up into several sentences. This is just one example.

We appreciate this comment pointing out that this part needs better structuring – as also mentioned by reviewer 1. Hence, we have introduced several subheadings to break up the complex content hoping to make it an easier read and also clearer to non-experts.

15) Section 5 states "Beyond CIN", but CIN has not been introduced before. In fact, there is not discussion of why cells should control centrosome numbers and maintain a single centrosome in cycling cells. How is this linked to their roles in mitosis and assembly of primary cilia?

We thank the reviewer for this comment that has also been pointed out by reviewer 3. We added a section on the interrelation between extra centrosomes, CIN and tumorigenesis with a particular focus on how CA provokes chromosome missegregation and CIN (on page 9; line 195).

16) The same section heading states "emerging roles of extra centrosomes". To me this sounds as if physiological functions are going to be discussed, but the section only deals with disease conditions. On page 8 the discussion switches to impaired centrosome structural integrity, which does not fit with the above heading.

We apologize for the confusion. With 'beyond CIN - emerging roles of extra (which we changed to amplified) centrosomes' we aim to discuss features of amplified centrosomes in cancer cells that act beyond mitotic processes, which have been shown to lead to CIN. We focus on interphase-specific processes, such as cell motility, invasion, lysosome function and organelle positioning. To make this clearer, we also changed the title of the section to 'emerging roles of centrosome aberrations in cancer cells'. We also include subheadings for clarity.

Similarly, the following section 6 heading states to discuss extra centrosomes in normal physiology, but begins with a discussion of abnormal centriole structure and PCM composition. Then, rather than extra centrosomes, loss of centrosomes is discussed.

We agree with the reviewer that the title of this section was misleading. We changed it to 'Modulation of centrosome numbers in physiology' to make clear that we don't talk exclusively about amplified centrosomes, but also other variations such as centrosome loss during differentiation or development.

17) The authors state several times that centrosomes "dissolve" in cardiomyocytes, please revise this term.

We apologize for this unconcise language, as pointed out also by reviewer 1. We replaced the term 'centrosome dissolution' by 'loss of centriole cohesion' as the work of Zebrowski et al demonstrate that centrioles split shortly after birth leading to loss of centrosome integrity. Centriole splitting correlates with relocalization of pericentrin and CDK5RAP2 to the nuclear membrane, both of which have been shown to be required for centrosome cohesion. We added this piece of information on page 13; line 454.

18) Sometimes the authors talk about extra centrioles and sometimes about extra centrosomes (e.g. lines 344 and 346, but also elsewhere). This is clearly not the same thing, but what is the difference and why does it matter? These issues need more precise language and explanation for non-expert readers.

We thank the reviewer for this notice and corrected 'centrosome' to 'centriole' in lines 344 and 346 (now lines 468 and 469 on page 14). In section 2 we provide a more detailed explanation on the structure/function of centrioles and centrosomes (page 4; line 98).

19) Figures: Overall, I found the figures to be quite disconnected from the text. Some parts are not discussed at all.

The structure of Figure 1 is not replicated in the text, e.g. there is no real discussion of centriole loss and structural defects and the related pathologies ciliopathy and microcephaly. Also, the cartoons used here display normal structures rather than diseased conditions (compared to the cartoons on the right). Similarly, under "physiology" some terms on the left simply name cell types rather than physiological processes. It is unclear what exactly is shown here.

We agree with the reviewer on that issue, as due to space limits, we initially aimed to fit important information into only three Figures, leading to loss of clarity. In the revised version we now provide 4 Figures and aligned their content better to the written part. For example, we now added a paragraph on the inflammasome in the revised version of the manuscript, which was absent in our initial submission (page 8; line 244).

Figure 2 displays signaling "downstream of extra centrosomes", but an extra centrosome is shown only in A. Is this on purpose? Does this apply only to having one extra centrosome as depicted in the cartoons? Does the number matter? I have not found any discussion of panel B in the text.

We now added a paragraph about centrosome-mediated inflammasome activation in immune cells in the main text to better explain the content of Figure 2 and the analogies with the PIDDosome complex (page 8; line 244).

Figure 3 shows non-canonical functions of centrosomes. Again, I did not find a discussion of this in the text. This figure is cited in the context of extra centrosomes. "Increased PCM and microtubules" in B refers to an increased canonical function, but is not a non-canonical function.

We now split former Figure 3 into two Figures (3 and 4) and focus only on emerging function of amplified centrosomes in cancer in Figure 3 and physiological functions of centrosome modifications (loss and amplification) in Figure 4.

Referee #3:

This review article discusses the consequences of extra centrosomes in pathological and physiological conditions, highlighting how these extra centrosomes are not just deleterious, but can be part of physiological process.

Overall, the review is very pleasant to read, presents an in depth and balanced discussion on the topic and its subject is timely. It does not just present a list of findings, but highlights a nice alternative to the prevalent hypothesis that extra centrosomes are mostly a dangerous and a pathological condition.

We thank Patrick Meraldi for the detailed evaluation of our manuscript and the positive feedback. We highly appreciate your knowledge and expertise in the field and tried to incorporate your suggestions and recommendations as good as possible. We feel that the manuscript greatly benefited from these modifications.

I have a number of minor concerns/comments that are listed below. The authors should consider those as suggestions that could further improve this review article:

- On page 2 and page 7 the authors allude to the role of extra centrosomes in generating chromosomal instability, but never present the mechanistic detail. Stating that extra centrosomes favor the formation of transient multipolar spindles that increase the proportion of erroneous kinetochore microtubule attachments that persist as cells enter anaphase (Ganem et al., 2009 but also Silkworth et al., 2009 that appeared at same time), would be helpful.

We thank you for this valid comment. We added a section on the interrelation between extra centrosomes, CIN and tumorigenesis with a particular focus on how CA provokes chromosome missegregation and CIN (on page 9; line 195) and added the suggested references.

- bottom of page 3: The authors might consider discussing briefly other structures contributing to microtubule nucleation in the mitotic spindle, such as the augmin pathway (work of the Gerlich laboratory) and kinetochore-driven microtubule nucleation (Maiato et al., 2004 and the

recent work of the Kops laboratory showing that this MT nucleation at kinetochores depends also on pericentrin)

*We included a paragraph on **page 4; line 118**, in which we discuss alternative structures, that have been shown to act as MT organizing sites in cells and included work from the Goshima and Gerlich lab about the Augmin complex and Augmin accumulation on long-lived MTs - as also suggested by reviewer 2 - the Khodjakov lab on kinetochore-driven MT nucleation and a recent review from Sabine Petry on branching MT nucleation.*

- page 4: When discussing the dissolution of the linker between the two centrosomes the author could mention the key role of the Nek kinases (work of the Fry and Roig laboratories) *We added a side-sentence on the regulation of centrosome disjunction by Nek2 kinase when we talk about dissolution of the centrosomal linker and included the respective references (**page 5; line 141**).*

- page 5: when discussing the coordination of DNA replication and centrosome duplication at the G1/S boundary, authors could cite the initial studies linking centrosome duplication to Cdk2 activity (Hinchcliffe et al., 1999; Meraldi et al., 1999 (I apologize for self-citing); Lacey et al., 1999 and Matsumoto et al., 1999).

*We apologize for not citing the original literature in a comprehensive manner and now added the missing references in the revised version of our manuscript (**page 6; line 178**).*

- page 5: when presenting mechanisms leading to extra centrosomes, the authors might want to also include the concept of breakage of excessively long centrioles, first proposed by the Bettencourt laboratory (Marteil et al., 2018)

*We thank you for that valuable suggestion. We included the concept of breakage of overly long centrioles as described by the Bettencourt-Dias lab on **page 6; line 200**.*

- page 5: when discussing how chromosome segregation errors and an abrogated mitosis can lead to extra centrosomes the authors might also include the original study describing such events (Meraldi et al., 2002)

*We added the reference on **page 7; line 121**, when we discuss mitotic defects as cause for numerical centrosome aberrations.*

- page 6: when talking about the mitotic surveillance pathway the authors might include a review article on the topic such as Phan and Holland 2021.

*We included the suggested review by Phan & Holland about extra centrosomes and the mitotic surveillance pathways on **page 7; line 223**.*

- page 7: when discussing extra centrosomes correlating with cancer grades the authors should again cite Marteil et al. 2018

*We added the reference on **page 10; line 318** when we refer to the correlation of centrosome numbers and metastatic outgrowth.*

- on page 10: when discussing the unknown role of extra centrosomes in the liver the authors might want to present the work of Duncan et al., Nature, 2010, which indicated that extra centrosomes can lead to multipolar spindles that favor a reduction in ploidy.

We thank you for this comment. Indeed, multipolar mitoses have been reported by the Grompe and Duncan labs as a means to restore diploidy in hepatocytes. This process can be seen as a means to reduce centrosome numbers, but only does so in the context of ploidy reduction, which is different from what we had in mind discussing as means to regulate centrosome counts.

Prof. Eva Kiermaier
Life&Medical Sciences Institute
Carl-Troll-Straße 31
Bonn, North Rhine Westphalia 53115
Germany

Dear Eva,

This is to briefly inform you that your manuscript has been accepted for publication in EMBO Reports. Your manuscript will be processed for publication by EMBO Press. It will be copy edited and you will receive page proofs prior to publication. Please note that you will be contacted by Springer Nature Author Services to complete licensing information.

Yours sincerely,

Deniz

--

Deniz Senyilmaz Tiebe, PhD
Senior Scientific Editor
EMBO Reports
